# Genetic dissection of the Transcription Factor code controlling serial specification of muscle identities in *Drosophila*

Laurence Dubois[1,2]*, Jean-Louis Frendo[1,2], Hélène Chanut-Delalande[1,2], Michèle Crozatier[1,2], Alain Vincent[1,2]*

[1]Centre de Biologie du Développement (CBD), CNRS and Université de Toulouse, Toulouse, France; [2]Centre de Biologie Intégrative (CBI), CNRS and Université de Toulouse, Toulouse, France

**Abstract** Each *Drosophila* muscle is seeded by one Founder Cell issued from terminal division of a Progenitor Cell (PC). Muscle identity reflects the expression by each PC of a specific combination of identity Transcription Factors (iTFs). Sequential emergence of several PCs at the same position raised the question of how developmental time controlled muscle identity. Here, we identified roles of Anterior Open and ETS domain lacking in controlling PC birth time and Eyes absent, No Ocelli, and Sine oculis in specifying PC identity. The windows of transcription of these and other TFs in wild type and mutant embryos, revealed a cascade of regulation integrating time and space, feed-forward loops and use of alternative transcription start sites. These data provide a dynamic view of the transcriptional control of muscle identity in *Drosophila* and an extended framework for studying interactions between general myogenic factors and iTFs in evolutionary diversification of muscle shapes.

**\*For correspondence:** laurence.dubois@univ-tlse3.fr (LD); alain.vincent@univ-tlse3.fr (AV)

**Competing interests:** The authors declare that no competing interests exist.

## Introduction

The morphological diversity of body wall muscles is necessary for precision, strength and coordination of body movements specific to each animal species. The development of the complex architecture of the body wall musculature of the *Drosophila* larva – 30 different muscles in each hemisegment (*Bate, 1993*) – is a classical model to decrypt transcription regulatory networks controlling muscle morphological diversity. Each muscle is a single multinucleated fiber built by fusion of a Founder Cell (FC) with fusion competent myoblasts (FCMs). Muscle identity - orientation, shape, size, attachment sites - reflects the expression by each FC of a specific combination of identity Transcription Factors (iTFs). Establishment of the FC iTF code starts with activation of specific muscle iTFs, in response to positional information from the ectoderm which defines equivalence groups of myoblasts within each segment, called promuscular clusters (PMCs) (*Carmena et al., 1995*; *Baylies et al., 1998*). The second step is the selection of progenitor cells (PCs) from each PMC, via interplay between Ras signaling and Notch (N)/Delta-mediated lateral inhibition, the unselected myoblasts becoming FCMs (*Carmena et al., 2002*). The third step is the asymmetric division of each PC into two FCs or, in some cases, one FC and one adult muscle precursor cell (AMP) or pericardial cell. Asymmetric division leads to maintaining expression of some iTFs in one FC and their repression by N signaling in the sibling cell, thereby contributing to muscle lineage diversity (*Ruiz-Gomez et al., 1997*; *Carmena et al., 1998*). This henceforth classical, three-step model of muscle identity specification relies heavily on positional information conferring each muscle its identity (*Tixier et al., 2010*). Interestingly, pioneering studies showed that specification of two nearby Even-

**eLife digest** Animals have many different muscles of various shapes and sizes that are suited to specific tasks and behaviors. The fruit fly known as *Drosophila* has a fairly simple musculature, which makes it an ideal model animal to investigate how different muscles form.

In fruit fly embryos, cells called progenitor cells divide to produce the cells that will go on to form the different muscles. Proteins called identity Transcription Factors are present in progenitor cells. Different combinations of identity Transcription Factors can switch certain genes on or off to control the muscle shapes in specific areas of an embryo. However, progenitor cells born in the same area but at different times display different patterns of identity Transcription Factors; this suggests that timing also influences the orientation, shape and size of a developing muscle, also known as muscle identity.

Dubois et al. used a genetic screen to look for identity Transcription Factors and the roles these proteins play in muscle formation in fruit flies. Tracking the activity of these proteins revealed a precise timeline for specifying muscle identity. This timeline involves cascades of different identity Transcription Factors accumulating in the cells, which act to make sure that distinct muscle shapes are made. In flies with specific mutations, the timing of these events is disrupted, which results in muscles forming with different shapes to those seen in normal flies. The findings of Dubois et al. suggest that the timing of when particular progenitor cells form, as well as their location in the embryo, contribute to determine the shapes of muscles.

The next step following on from this work is to use video-microscopy to track identity Transcription Factors when the final muscle shapes emerge. Further experiments will investigate how identity Transcription Factors work together with proteins that are directly involved in muscle development.

skipped (Eve) expressing PCs was sequential (*Buff et al., 1998*; *Halfon et al., 2000*), but the link between PC birth time and muscle identity remained to be explored.

We have previously shown that four PCs, at the origin of one dorsal muscle (DA2) and one AMP, and the 6 dorso-lateral (DL) muscles, DA3, DO3, DO4, DO5, DT1, and LL1, are serially selected from a PMC expressing Collier (Col/Kn, Early B-Cell Factor (EBF) in vertebrates (*Daburon et al., 2008*). More precisely, the DA2/AMP, DA3/DO5 and LL1/DO4 PCs are sequentially selected at roughly identical positions in thoracic and abdominal segments, while the DO3/DT1 PC is selected at a slightly posterior position and only in abdominal segments (*Boukhatmi et al., 2012*; *Enriquez et al., 2012*; *Figure 1A*). Beyond the PC step, *col* transcription is only maintained in the DA3 muscle (*Crozatier and Vincent, 1999*), while other iTFs, the C2H2 zinc finger protein Krüppel (Kr), the homeodomain protein S59 (vertebrate NKx1.1) and the Lim-homeodomain protein Tailup (Tup/ Islet1) are expressed in the LL1, DT1 and DA2 lineages, respectively (*Dohrmann et al., 1990*; *Ruiz Gomez and Bate, 1997*; *Boukhatmi et al., 2012*). Serial emergence of DL PCs, followed by lineage-specific expression of different iTFs, raised the question of how PC selection timing and muscle identity were linked. The discovery that Tup expression led to *col* repression in the DA2/ AMP PC, thereby distinguishing between DA2 and DA3 identities, provided a first insight into this question. We indeed found that the time lag between DA2/AMP and DA3/DO5 PC selection coincides with the period of dorsal regression of Tinman (Tin; vertebrate Nkx2.5) expression (*Johnson et al., 2011*), such that only the first-born DA2 PC inherits Tin levels above the threshold required for activation of *tup* and imposing a DA2 fate. Yet, our understanding of how conjunction of developmental time and position translates into muscle-specific iTF codes, remained fragmentary.

Here, we identified several new muscle TFs, starting from a systematic deficiency screen of the second chromosome, i.e., roughly 40% of the *Drosophila* genome. We describe the roles of No Ocelli (Noc), a NET family zinc finger protein (*Cheah et al., 1994*), Sine oculis (So), a member of the Six family of homeodomain proteins (*Cheyette et al., 1994*; *Serikaku and O'Tousa, 1994*; *Kenyon et al., 2005*), and the co-factor ETS domain lacking Edl (*Baker et al., 2001*; *Yamada et al., 2003*; *Qiao et al., 2006*) in DL muscle development, and in more detail, roles of Eyes-absent (Eya), a partner of Six proteins (*Pignoni et al., 1997*), and Anterior open (Aop), an Ets-domain transcription

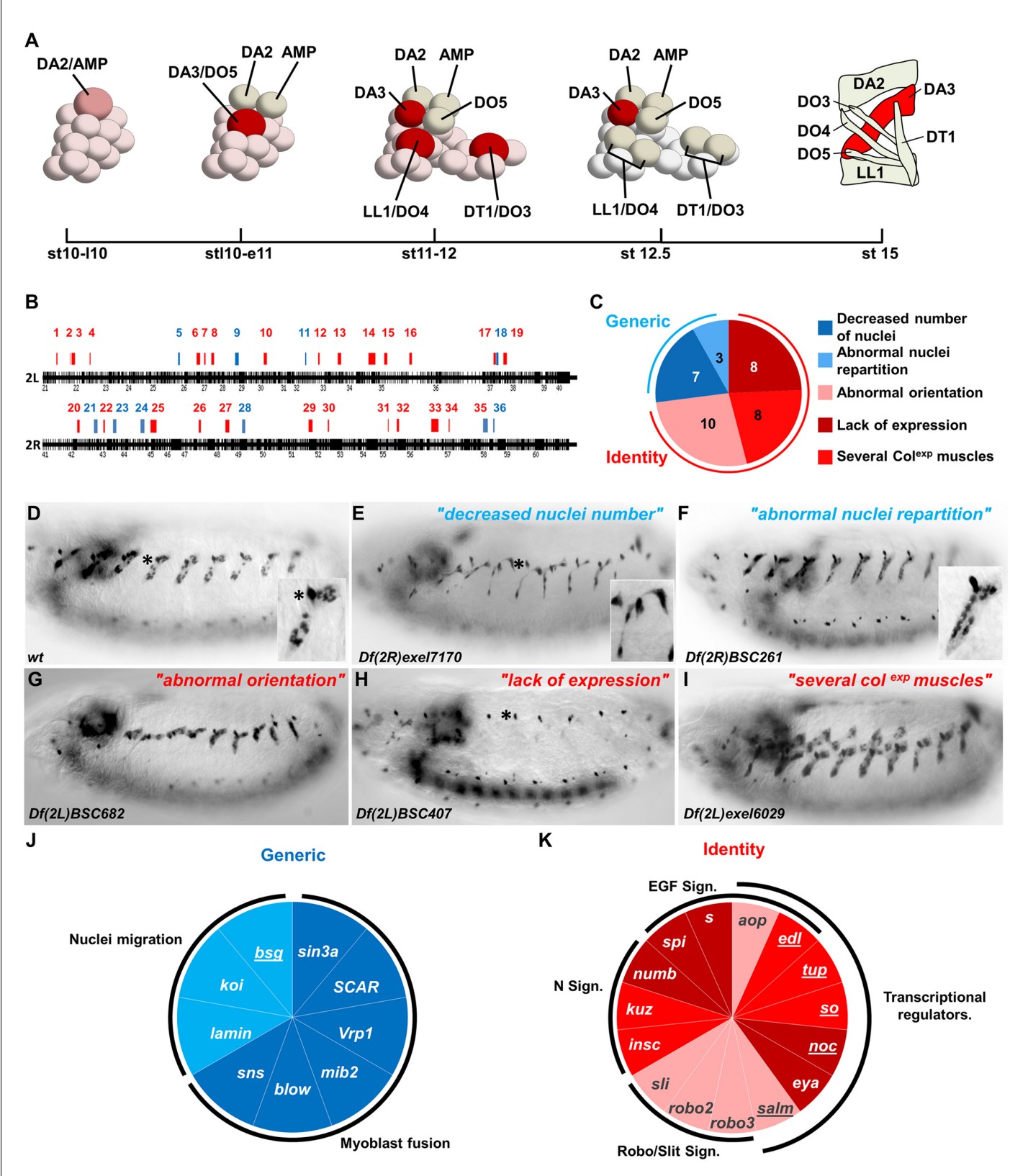

**Figure 1.** Genetic identification of muscle identity genes. (**A**) Diagrammatic representation of the sequential emergence of four PCs (large cells) from the Col expressing PMC, followed by PC into FC divisions (embryonic stages (st) 10–12.5) and the corresponding muscle pattern at stage 15. The name
*Figure 1 continued on next page*

*Figure 1 continued*

of each PC, FC and muscle is indicated. Col expression is in red, color intensity indicating expression level. (B) Gene density along chromosome 2L and 2R, schematized by black bars. Position and size of each of 36 regions identified in our screen are indicated by red or blue bars. (C) Pie chart showing repartition of the DA3 phenotypes into two classes of generic (blue), and identity (red) defects. (D–I) Col immunostaining of late stage 15 embryos; (D) wt and (E–I), representative examples (deficiency name indicated) of each phenotypic class. The asterisk in (D,E,H) labels a dorsal class IV md neuron expressing Col. In this, and following figures, lateral views of embryos are shown, anterior to the left. (J,K) Pie charts associating individual genes with generic myogenic (K) or identity (L) mutant phenotypes. See also *Figure 1—source data 1* for phenotypes.

The following source data and figure supplement are available for figure 1:

**Source data 1.** 36 chromosomal deficiencies showing DA3 muscle phenotypes.

**Figure supplement 1.** EGF-R signaling is required for a normal pattern of DL muscles.

repressor (*Rebay and Rubin, 1995*; *Xu et al., 2000*). Analysis of the *aop, edl, eya, noc* and *so* muscle mutant phenotypes and time windows of transcription, combined with *col* transcription in the different mutant contexts, revealed a cascade of regulations including coherent and incoherent feedforward loops, which link PC selection time to muscle identity. *aop* and *edl* control the temporal sequence of DL PC selections, *eya* is required in PCs for maintaining iTF transcription, while *so* and one *eya*-specific isoform are deployed at the FC step. Finally, *noc* regulates expression of other iTFs, at the PC or FC step, depending upon the muscle lineage. Integration of these new data with pre-existing knowledge provides a comprehensive, dynamic view of the transcriptional control of muscle identity in *Drosophila*, and an extended framework for studies of interactions between general myogenic factors such as Nautilus (Nau)/MyoD and Eya, and iTFs in the diversification of muscle lineages during animal evolution.

## Results

### A genetic screen for muscle defects

In order to identify new muscle identity genes, we screened a collection of 389 overlapping deficiencies, each deleting between 10 and 15 genes, and together covering about 80% of the *Drosophila* second chromosome (*Chanut-Delalande et al., 2014*). Homozygous deficiency embryos were first examined for DA3 Col expression at the end of the fusion phase, embryonic stage 15. Nuclear Col localization allowed appraisal both of DA3 formation and shape, the number and spatial distribution of DA3 nuclei, and the presence of ectopic Col-expressing muscles. General embryonic defects could be identified by the loss, or gross disturbance of Col expression elsewhere, in the central and peripheral nervous systems, and/or lymph gland (*Dubois and Vincent, 2001*), and the corresponding chromosomal deficiencies were not considered here. 36 were retained (*Figure 1B* and *Figure 1—source data 1*). The observed DA3 phenotypes were divided into two broad classes (*Figure 1C*): Class 1: Decreased number or abnormal repartition of nuclei (*Figure 1E–F* compare to *Figure 1D*); Class 2: Abnormal DA3 orientation and/or either loss of Col expression or ectopic Col expression in additional muscles (*Figure 1G–I* compare to *Figure 1D*). Three deletions showing both DA3 abnormal orientation and low nuclei number were considered as class 2 (regions 3, 4 and 30, *Figure 1—source data 1*). Class 1 phenotypes have previously been observed in myoblast fusion or nuclei migration mutants which affect roughly equally all muscles (*Folker et al., 2014*; *Rushton et al., 1995*) and were considered here as 'generic myogenesis' defects (*Figure 1C* and *Figure 1—source data 1*). Class 2 phenotypes were reminiscent of either iTF or *Notch (N)* mutants (*Ruiz-Gomez et al., 1997*; *Crozatier and Vincent, 1999*; *Tixier et al., 2010*) and considered as 'muscle identity' defects (*Figure 1C* and *Figure 1—source data 1*).

To identify the gene(s) whose loss caused a DA3 phenotype in mapped deletions, we tested the most promising candidates for which loss of function mutants were available. Genes for which mutants over the deficiency reproduced the deficiency phenotype were selected for further analysis. From a total of 36 different chromosomal regions, we identified 9 genes out of 10 regions linked to generic defects and 15 genes in 14 regions linked to identity defects (*Figure 1J,K* and *Figure 1—source data 1*). The relevant gene(s) in 12 other regions remain to be identified. Seven of the nine

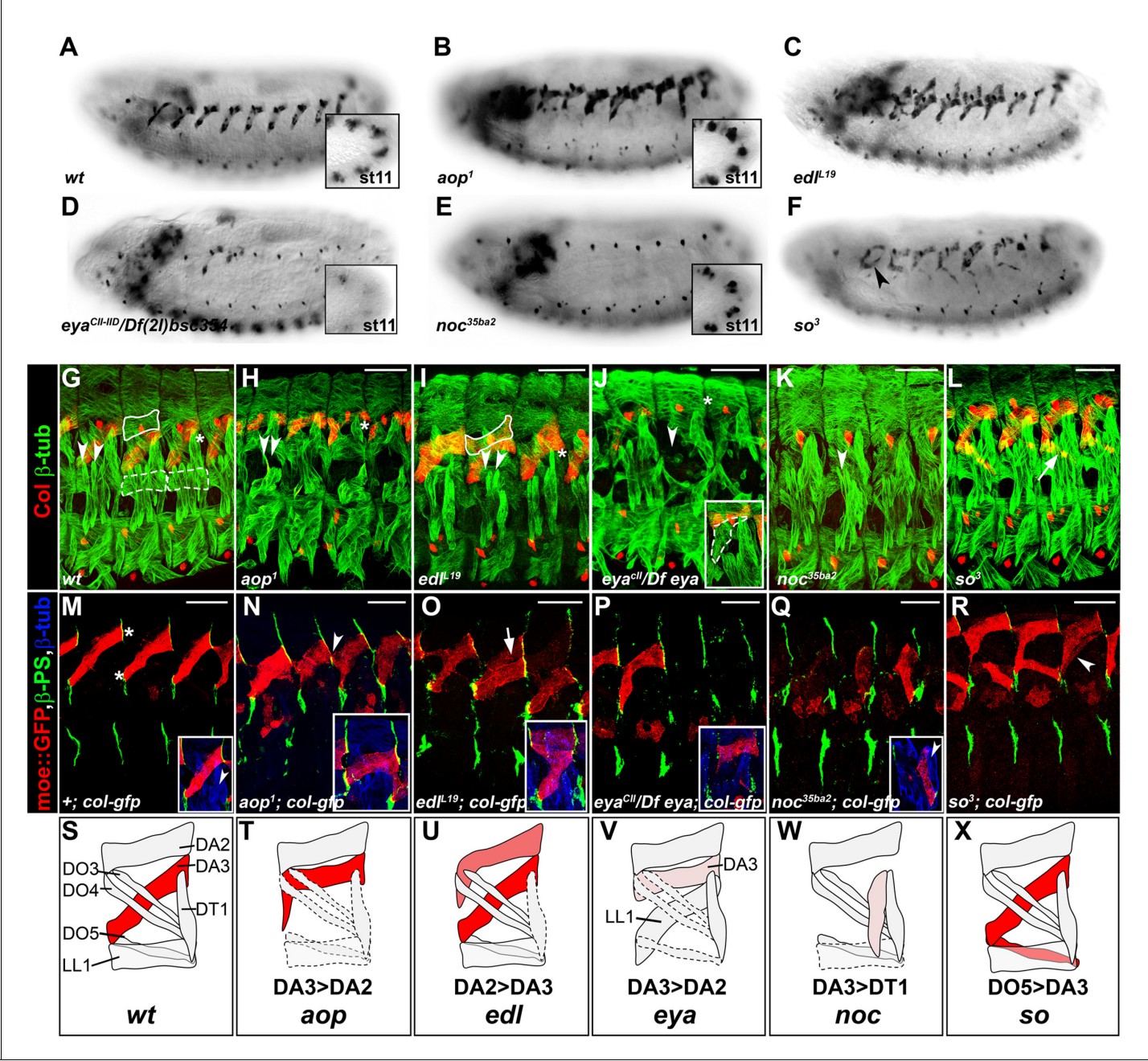

**Figure 2.** Specific muscle patterning defects in *aop, edl, eya, noc* and *so* mutant embryos. (A–F) Late stage 15 embryos stained for Col, to visualize the DA3 muscle. (A) wt, (B–F) embryos homozygous mutant for *aop, edl, eya, noc* and *so* null alleles with their names indicated. Inserts in (A,B,D,E) show Col expression in PCs, stage 11. (G–L) stage 16 embryos stained for Col (red) and β3-tubulin (green) to visualize all body wall muscles; arrowheads point to LT1 and LT2, asterisks indicate DT1; DA2 is surrounded by a line in G,I, and LL1 by a dotted line in G. (G) wt embryo. (H) *aop¹* (I) *edl¹⁹*; Col is expressed in DA2 and DA3. (J) *eyaᶜⁱⁱ/Df(2L)ᴮˢᶜ³⁵⁴*; DA3 Col expression is lost; inset, LL1>DA3 transformation. (K) *noc³⁵ᵇᵃ²*: Col expression is lost. (L) *so³*; Col expression in DO5 (arrow). (M–R) Stage 16 embryos stained for β3-tubulin (blue), βPS integrin (green), to visualize tendon cell-muscle connections and moeGFP (red) expressed under control of a DA3-specific *col* CRM (*colᴸᶜᴿᴹ*), abbreviated *col-gfp*. Only βPS integrin and moeGFP are shown, except insets. (M) wt; DA3 ventral and dorsal attachment along the anterior and posterior segmental borders, respectively, are indicated by asterisks; inset, DT1 (arrowhead). (N) *aop*; DA3 with both DA3 and DA2-like (arrowhead) anterior attachments; DA3>DA2 transformation, inset. (O) *edl*: moeGFP expression in DA2 and DA3 (arrow and inset). The arrow indicates partial DA2>DA3 transformation; inset, bifid anterior DA3 attachment. (P) *eya*: moeGFP is lost in most segments or indicates partial or complete (inset) DA3>DA2 transformation. (Q) *noc*: DA3>DT1 transformation, resulting in DT1 (arrowhead in inset) duplication. (R) *so*: moeGFP expression in DO5; DO5>DA3 transformation (arrowhead) in some segments. (S–X) Schematic diagram

*Figure 2 continued on next page*

*Figure 2 continued*

of the most frequent DA2 and DL muscle phenotypes in *aop, edl, eya, noc* and *so* mutants; Col expression is in red; see *Figure 2—source data 1* for statistics. Bars: 30 µm

The following source data and figure supplements are available for figure 2:

**Source data 1.** Quantification of muscle phenotypes observed in *aop, edl, eya, noc* and *so* mutant stage 15 embryos.

**Figure supplement 1.** *salm[1]* is required for proper skeletal attachment and morphology of the DA3 muscle.

**Figure supplement 2.** Snapshots for *Videos 1–6*.

genes in the generic class encode cytoskeletal or membrane-associated proteins with an already well-known role in either myoblast fusion or nuclei repartition in muscle syncitia, validating our screen (*Figure 1J*; *Kim et al., 2015*). The eigth gene is *sin3A*, a chromatin binding protein present in transcription repressor complexes, also required for a normal pattern of myoblast fusions (*Dobi et al., 2014*). The 9[th] gene is *basigin* (*bsg*), a predicted igG family plasma membrane protein, interacting with integrin (*Curtin et al., 2005*), whose role in muscle development has not been assessed.

Among the 15 genes associated with identity phenotypes (*Figure 1K*), six encoded components of either the N or Robo/Slit signaling pathways, two pathways previously implicated at different steps of DA3 muscle formation (*Crozatier and Vincent, 1999*; *Ordan et al., 2015*) and were therefore not further studied. Four other encoded components of the EGF-R signaling pathway: *spitz (spi), Star (S), aop* (also called *yan;* Flybase FBgn 000097), and *edl* (also called *mae;* Flybase FBgn0023214), while a deficiency (Df(2R)BSC259) removing both mesodermal FGFs, *Thisbe* and *Pyramus*, (*Stathopoulos et al., 2004*) did not show a DA3 phenotype. A complete lack of DA3, DO5, DO4 and LL1 muscles in mutants for either *spi (spi[IIA])*, the EGF-R signal, or *Star (S[IIN])*, a chaperone protein required for Spi processing (*Heberlein and Rubin, 1991*), confirmed the central role of Epidermal Growth Factor-Receptor (EGF-R) signaling in specification of these DL muscles (*Figure 1—figure supplement 1*).

## New transcriptional regulators involved in muscle identity

Seven identity genes encoded transcriptional regulators, and potentially, new muscle iTFs: *aop, edl, eya, noc, so, spalt major (salm)* and *tup* (*Figures 1K* and *2A–F*). Previous studies showed that the Ets-domain transcription activator Pointed (Pnt), and transcription repressor Aop/Yan (*Xu et al., 2000*; *Rebay and Rubin, 1995*), promoted and inhibited the formation of Eve-expressing dorsal PCs, respectively, downstream of EGF-R signaling (*Halfon et al., 2000*; *Carmena et al., 2002*). The mesodermal function of *edl* remained, however, unknown. Comparing the *aop* and *edl* phenotypes thus provided an opportunity to further characterize outputs of EGF-R signaling in muscle identity specification. Whereas we previously reported *tup* function in dorsal muscle identity, neither function

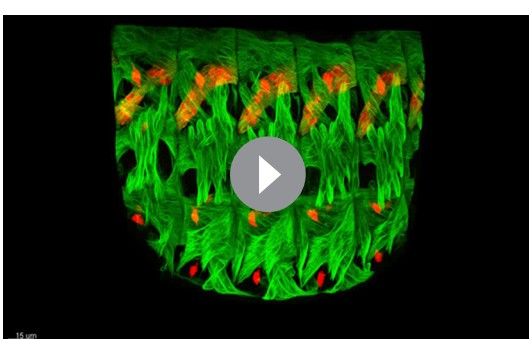

**Video 1.** 3-D view of the muscle pattern in stage 16 wt embryos, *Figure 2G*.

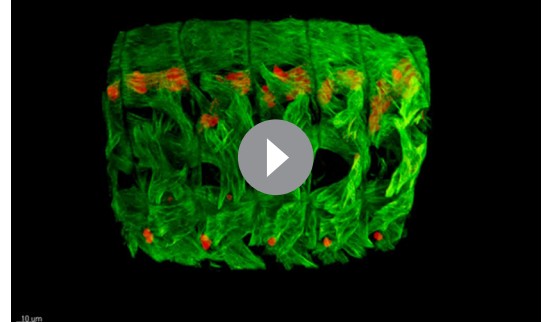

**Video 2.** 3-D view of the muscle pattern in stage 16 *aop[1]* embryos, *Figure 2H*.

of *noc, salm* nor *so* in muscle development was previously characterized. *Drosophila* Six4 and So are orthologous to Six proteins which interact with Eya in mouse myogenic progenitors (*Heanue et al., 1999*; *Relaix et al., 2013*). *eya* was proposed to interact with *Six4* in *Drosophila* somatic muscle development, both genes showing similar expression patterns (*Clark et al., 2006*; *Liu et al., 2009*). Our identification of *so* mutants in our screen and the difference between the *eya* and *so* phenotypes (*Figure 1—source data 1*) called for a detailed comparison of *eya* and *so* expression and function in muscle PCs.

## *aop, edl, eya, noc* and *so* muscle phenotypes

To better assess the muscle phenotypes associated with each mutant, we examined the pattern of DL muscles in late stage 15 embryos immunostained for β3Tubulin, using Col staining to visualize DA3 (*Figure 2G–L*, *Video 1*). In addition, we introduced the DA3-specific *col^{LCRM}*-moeGFP reporter gene (*Enriquez et al., 2012*), to precisely visualize the DA3 contours in these mutant backgrounds. Stability of the MoeGFP fusion protein also allowed following 'DA3' muscles in *noc* and *eya* mutants which lack Col expression (*Figure 2M–R*). In order to verify that muscle phenotypes were not associated with defective tendon cell differentiation, we stained mutant embryos for βPS integrin which accumulates at muscle-tendon junctions (*Leptin et al., 1989*). Based on this analysis, we eliminated *salm* (*salm^1*) mutants which exhibited a phenotype reminiscent of defective tendon cells (*Schnorrer and Dickson, 2004*; *Staudt et al., 2005*; *Figure 2—figure supplement 1*). Unlike *salm*, no βPS integrin accumulation defects were detected in null mutants for *aop* (*aop^1*), *edl* (*edl^{L19}*), *noc* (*noc^{35ba2}*), *eya* (*eya^{CII-IID}*), and *so* (*so^3*) mutants (*Figure 2M–R*), confirming muscle identity defects.

In *aop* mutants, the DA3 muscle(s) was misshapen in 2/3 of segments (n = 123), with cases of DA3 to DA2 transformation (DA3>DA2; *Figure 2B*, *Figure 2— figure supplement 2*, *Video 2* and *Figure 2—source data 1*). The DT1 and LL1 muscles were also malformed in 40% segments (in 48/123 and 47/123 segments, respectively) and lateral and ventral muscles were severely disorganized. The DA2 muscle was unaffected (*Figure 2H*—and *Figure 2—source data 1*). *col^{LCRM}*-moeGFP expression confirmed an abnormal shape of the DA3 muscle suggestive of partial DA3>DA2 transformation (*Figure 2N,T*). In *edl* mutants, a second Col-expressing muscle was observed in some segments, sometimes associated with morphological change suggestive of DA2>DA3 transformation (*Figure 2C,I*, *Figure 2— figure supplement 2*, *Video 3* and *Figure 2—source data 1*). The DT1 muscle was either absent or too de-structured to be assigned specific identities (in 61/124 segments; *Figure 2I*). *col^{LCRM}*-moeGFP expression confirmed a DA2>DA3 transformation in 47% of segments (*Figure 2O*), but also revealed a number of reciprocal at least partial DA3>DA2 transformations (16/124 segments) (*Figure 2O* inset—and *Figure 2—source data 1*). In *eya* mutants, DA3 Col expression was lost, a loss already observed at the PC stage (*Figure 2D,J*). Consistent with loss of Col expression early during muscle specification, the DL muscle pattern was severely disorganized in most of segments. The LL1 was absent or oriented like DA3 (in 129/140 segments; *Figure 2J*, *Figure 2— figure supplement 2*, *Video 4* and *Figure 2—source data 1*) a phenotype already observed in *col* mutant embryos (*Enriquez et al., 2012*). The LT muscles were also often missing and some ventral muscles were absent or malformed, as previously reported (*Figure 2J*; *Figure 2—source*

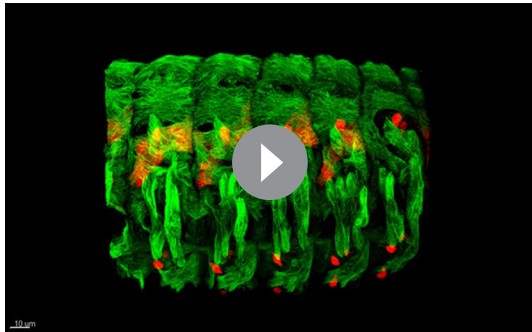

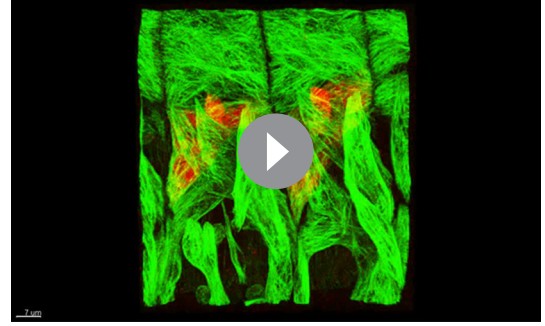

**Video 3.** 3-D view of the muscle pattern in stage 16 *edl^{L19}* embryos, *Figure 2I*.

**Video 4.** 3-D view of the muscle pattern in stage 16 *eya^{CII}/Df(2L)^{BSC354}* embryos, *Figure 2J*.

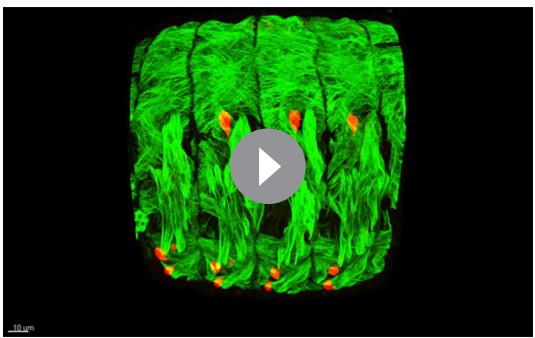

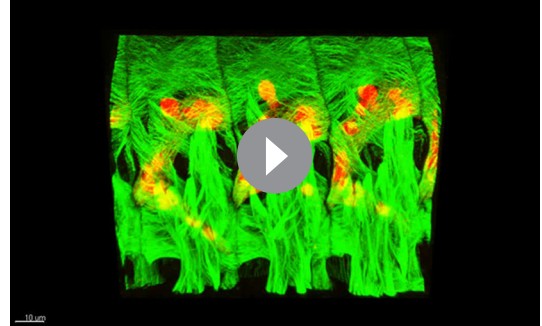

**Video 5.** 3-D view of the muscle pattern in stage 16 *noc35ba2* embryos, *Figure 2K*

**Video 6.** 3-D view of the muscle pattern in stage 16 *so3* embryos, *Figure 2L*

*data 1*) (*Liu et al., 2009*). Conversely, dorsal muscles and DT1 appeared normal (*Figure 2J*). *colLCRM-moeGFP* expression revealed a, sometimes complete or partial, DA3>DA2 transformation (*Figure 2P,V*). *noc* mutant embryos also lacked DA3 Col expression at stage 15 (*Figure 2E,K* and *Figure 2*, *Figure 2— figure supplement 2*, *Video 5* and *Figure 2—source data 1*). Contrary to *eya* mutants, however, Col expression was detected at the PC stage (insets in *Figure 2E*), indicating a role of *noc* in maintenance of Col expression in the DA3 lineage. *colLCRM-moeGFP* expression further revealed that the DA3 could orient like a DT1 in most segments, indicating a DA3>DT1 identity shift resulting in DT1 duplication (*Figure 2Q* and and *Figure 2—source data 1*). LT muscles were also affected (*Figure 2K*). In *so* mutants, Col ectopic expression was specifically observed in DO5 (*Figure 2F,L*, *Figure 2—figure supplement 2*, *Video 6*, and *Figure 2—source data 1*). *colLCRM-moeGFP* expression both confirmed *col* ectopic expression in the DO5 muscle and its DA3-like orientation in a fraction of segments (*Figure 2R*, arrowhead, indicating a partial DO5>DA3 identity shift in 79/116 segments (*Figure 2F,L* and *Figure 2—source data 1*).

In summary, we found that *aop, edl, eya, noc, and so* mutants display distinctive patterns of DL muscle defects and DA3 transformations, associated with modifications of Col expression (schematized in *Figure 2S–X*), indicating that each gene acts in different subsets of DL muscles, and/or at different steps of muscle identity specification.

## Nau/MyoD is transiently expressed in PMC cells subject to high EGF-R signaling

Understanding the specific muscle transformations observed in *aop, edl, eya, noc, and so* mutants required determining their expression patterns at the PMC, PC and FC stages. In order to access dynamic aspects of this expression, we used FISH with intronic probes which detect nascent transcripts and allow precisely determining temporal windows of transcription. To follow PC delamination events, we used Nau, the *Drosophila* ortholog of vertebrate myogenic regulatory factors (MRFs), a marker of PCs and FCs (*Michelson et al., 1990*; *Nose et al., 1998*). High-resolution 3-D analyses allow us to unambiguously identify the DA2/AMP and DL PCs and the derived DA2, DA3 and DO5 FCs and AMP (*Enriquez et al., 2012*; *Figure 1A* and *Figure 3— figure supplement 1*, *Video 7*). In early stage 10 embryos, the first selected, DA2/AMP PC is recognizable as a large apical cell, expressing high Nau levels (*Figure 3B,B'*). At stage 11, after the DA2/AMP PC has divided, the DA3/DO5 PC is observed, adjacent to the DA2 FC (*Figure 3C*). 3-D analyses also revealed previously undescribed, low level Nau expression in two or three cells surrounding each PC being selected (*Figure 3B,F*; *Figure 3— figure supplement 2*). This Nau expression

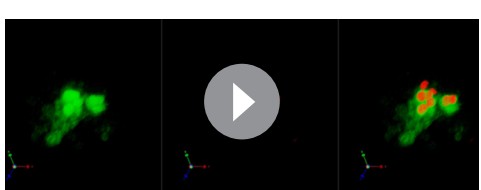

**Video 7.** Multiple FCs originate from the Col-expressing PMC.

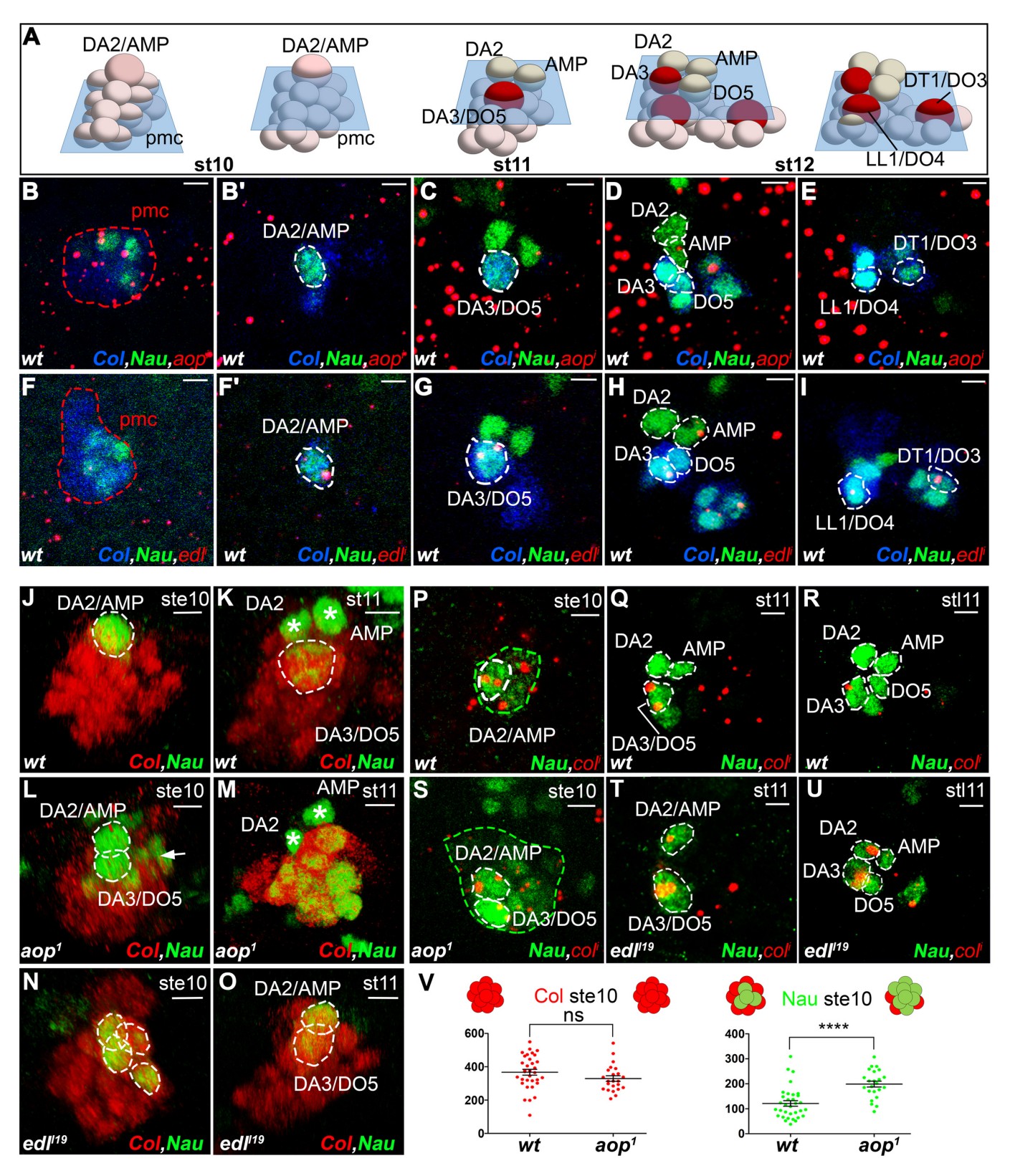

**Figure 3.** *aop* and *edl* differential expression and roles during PC selection. (**A**) Schematic representation of the positions of DL PCs and FCs, relative to the A/P, D/V and proximal/distal axes in stage 10, 11 and 12 wt embryos (*Video 7*); the blue trapeziums indicate planes of section shown in panels
*Figure 3 continued on next page*

*Figure 3 continued*

(B–K) and (R–U); Col expression is in red. (B–I) ISH to *aop* (B–E) and *edl* (F–I) primary transcripts (red dots), in wt embryos stained for Col (blue) and Nau (green), at stages indicated above; early stage is abbreviated ste; two different planes of the same embryo are shown in B and B', F and F'. *aop* transcription in the Col PMC (B), and the AMP (D). (F–I) *edl* transcription in all PCs, the AMP and the DA3 FC. (B,F) Nau accumulation in two to three Col PMC cells, below the emerging PC. (J,O) 3D reconstruction of the Col PMC during DA2/AMP and DA3/DO5 PC selection; Col staining, red, Nau, green. The embryonic stage indicated in each panel. (J, K) wt; (J) apical DA2/AMP PC (dotted white circle); (K) apical DA3/DO5 PC (dotted circle), DA2 FC and AMP (asterisks). (L, M) *aop* embryos; (L) premature DA3/DO5 PC selection; additional Nau-expressing PMC cells (arrow), become PCs, (M). (N, O) *edl* embryos; (N) no PC is selected; a group of 3 to 4 Nau-expressing cells is embedded in the Col PMC; (O) two PCs are simultaneously selected. (P–U) ISH to *col* primary transcripts (red), Nau staining (green); (P–R) wt; sequential *col* transcription in the PMC and DA2/AMP PC (P), DA3/DO5 PC (Q), and DA3 FC (R). (S) *aop* mutant: simultaneous *col* transcription in two apical PCs; increased number of low level Nau-expressing cells (green dotted circle). (T,U) *edl* mutant; ectopic col transcription in the DA2/AMP PC (T) and DA2 FC (U). (V) Measurement of the diameter of Col (red) and Nau (green) expressing domains in early stage 10 wt and *aop* embryos. The Col expression domain is identical (P value = 0,1410; ns) and Nau domain expanded in *aop* compared to wt (P value<0,0001; ****), schematized on top of the statistics. Bars: 5 μm

The following figure supplements are available for figure 3:

**Figure supplement 1.** Snapshot for *Video 7*.

**Figure supplement 2.** Transient Nau expression in subsets of PMC cells.

**Figure supplement 3.** Extended analysis of the *aop* muscle mutant phenotype.

pattern was reminiscent of subgroups of PMCs cells displaying higher level dpMAPK (di-phospho mitogen-associated protein kinase), diagnostic of EGFR activity (*Carmena et al., 1998*, *2002*) and postulated to be cells primed to become PCs. Double staining confirmed that the Nau and dpMAPK patterns overlap, revealing that low level Nau expression corresponds to cells transitioning from PMC to PC, before reaching high level in selected PCs (*Figure 3—figure supplement 2*). Using Nau staining allowed us to follow these cells and PCs in subsequent FISH experiments.

### *aop* and *edl* activities control sequential DA2/AMP and DA3/DO5 PC selection

FISH experiments indicated *aop* transcription in Col PMC cells (*Figure 3B*), but not PCs (*Figure 3B'–E*), indicating its repression during the PC selection process, consistent with a role in promoting FCM fate (*Carmena et al., 2002*) and being a target of the FCM TF Lameduck (*Ciglar et al., 2014*). Conversely, *edl* transcription was not detected in PMC cells, but in PCs and the dorsal AMP and DA3 FC (*Figure 3F–I*). Thus *aop* and *edl* show sequential, complementary transcription patterns. In stage 10 *aop* mutant embryos, *col* transcription was detected in PMC cells as in wt (*Figure 3P,S and V*). However, a significantly increased number of cells expressing low Nau level revealed that *aop* down-regulation of EGF-R signaling was required to restrict Nau expression to PMC cells primed to become PCs (*Figure 3P,S,V*). Furthermore, 3-D reconstructions confirmed the presence of two apical cells expressing high Nau Level (21/27 PMCs with 2, and 6/27 with 1 apical PC), when only the DA2/AMP PC was observed in wt (30/35 PMC with 1 apical PC and 5/35 with no apical PC; *Figure 3J,L*). Thus, concomitant, early selection of two PCs in *aop* embryos prefigures the DA3>DA2 transformation (*Figure 2T*). At stage 11, Nau remained expressed at high level in several cells, showing that supplementary PMC cells are primed to become PCs (*Figure 3M*, compare to). This corroborates the observation of several aligned DA3-like fibers revealed by phalloidin staining of stage 16 *aop* mutant embryos (*Figure 3—figure supplement 3*). Conversely, in *edl* mutant embryos, no high Nau-expressing apical cell was observed at stage 10 (30/37 segments), when the DA2/AMP PC is selected in wt embryos. Rather, three to four small low Nau-positive cells remained embedded in the Col PMC (*Figure 3N* compare with *Figure 3J*) (*Figure 3O* compare to *Figure 3K*) both of which transcribed *col* (*Figure 3T* compare with *Figure 3Q*). Accordingly, *col* transcription was maintained in two cells at late stage 11, at positions corresponding to DA2 and DA3 FCs in wt embryos (*Figure 3U* compare with *Figure 3R*). In summary, we found that sequential PC selection is inversely compromised in *aop* and *edl* mutants. In absence of *aop*, the DA3/DO5, and supernumerary PCs are selected early, and in absence of *edl*, the DA2/AMP is selected too late.

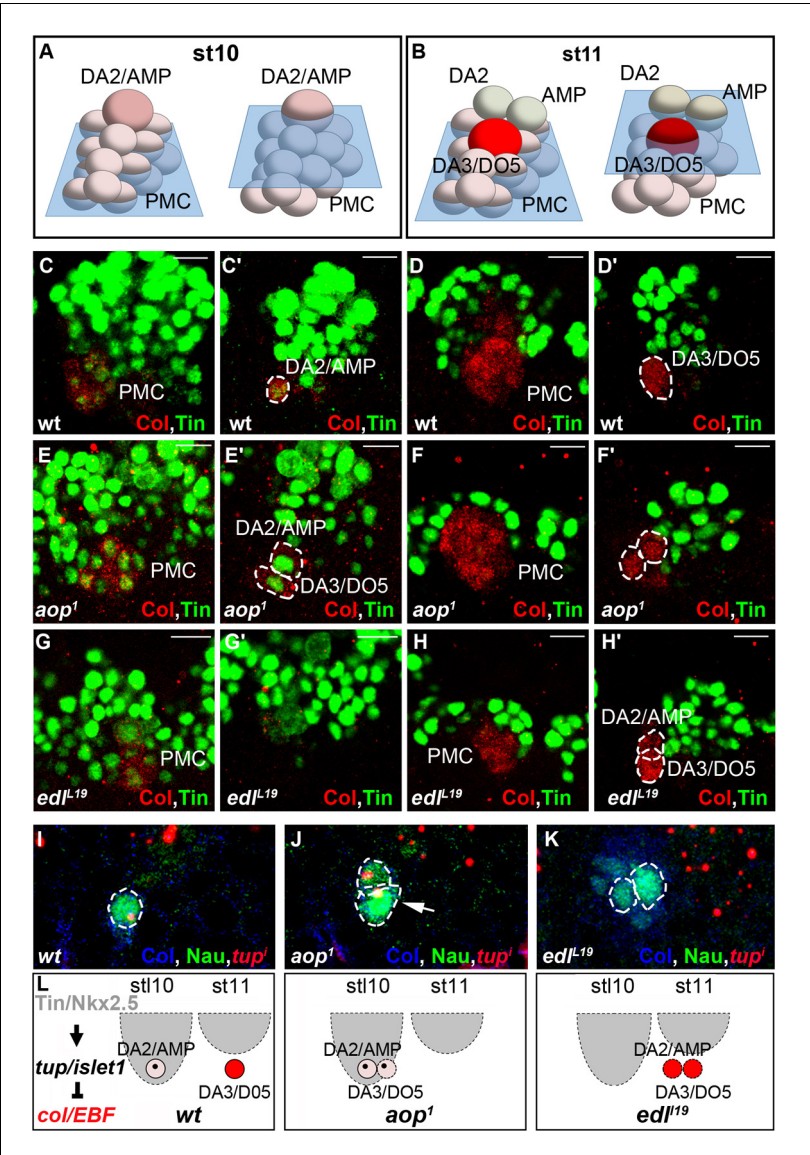

**Figure 4.** *aop* and *edl* control the temporal sequence of PC selection. (**A,B**) Schematic representation of the DA2/AMP PC and FCs and DA3/DO5 PC, at stages 10 (**A**) and 11 (**B**); the blue trapeziums indicate the planes of section shown below. (**C–H'**) Tin (green) and Col (red) embryo staining. (**C, C'**) Tin expression in the DA2/AMP PC and underlying PMC cells. (**D, D'**) Tin expression has regressed dorsally; the DA3/DO5 PC and underlying PMC cells are Tin negative. (**E–F'**) *aop* mutants; (**E, E'**), stage 10, two Col and Tin-expressing PCs are selected; (**F, F'**) stage 11, Col positive, Tin-negative cells are selected. (**G–H'**) *edl* mutants; (**G, G'**), No PC is selected. (**H, H'**) Two PCs are selected after dorsal regression of Tin expression. (**I–K**) FISH to *tup* primary transcripts in wt (**I**) *aop* (**J**) and *edl* (**K**) stage 10 (**I, J**) and 11 (**K**) embryos, stained for Col (blue) and Nau (green). In wt embryos (**I**) *tup* expression is only detected in the first selected Col positive PC (DA2/AMP, 100% n = 27). In *aop* mutants (**J**), *tup* transcription is sometimes detected (17% n = 34) in a second Col positive PC (arrow). In *edl* mutants (**K**), *tup* transcription is frequently lost in Col positive PCs (82% n = 29). (**L**) Summary scheme of the *aop* and *edl* phenotypes; wt, late stage (stl) 10; only the first-born, DA2/AMP PC inherits Tin, and activates *tup*, preventing Col autoregulation (left) which occurs in the second born, DA3/DO5 PC, in absence of Tin and Tup, stage 11 (***Boukhatmi et al., 2012***). In *edl* and *aop* mutants, the temporal sequence of PC selection is compromised; it occurs too early and too late in *aop* and *edl* mutants, respectively, leading to confusions of DA2 and DA3 fates. Bars: 10 µm

We have previously shown that the time lag between DA2/AMP and DA3/DO5 PC selection coincided with a period of dorsal regression of Tin expression, such that only the first selected, DA2/AMP PC inherited Tin (*Boukhatmi et al., 2012*; *Figure 4C–D'*). Tin staining of *aop* mutant embryos showed that the two apical cells observed at stage 10 inherit Tin (*Figure 4E,E'*), confirming advanced selection of the DA3/DO5 PC. Conversely, in *edl* mutants, none of the apical cells inherited Tin, confirming a delayed selection of the DA2/AMP PC (*Figure 4G,H'*), consistent with both transcribing *col* (*Figure 3T,U*). To verify that the observed shifts in PC selection timing lead to shifts of PC identity, we analyzed *tup* transcription. As previously shown, only the DA2/AMP PC inherits Tin levels above the threshold required for *tup* activation (*Figure 4I*), leading in turn to *col* repression and initiation of *tup* auto-regulation (*Boukhatmi et al., 2012*). We found that, in *aop* mutants, early selected PCs transcribed *tup* (*Figure 4J*), while in *edl* mutants, late selected PCs did not (*Figure 4K*), mirroring *col* transcription (*Figure 3T,U*). Together, Tin, *tup* and *col* expression data, and the DA3>DA2 and DA2>DA3 muscle transformations predominantly observed in *aop* and *edl* embryos, respectively, show that timely PC selection is essential for each PC to inherit different Tin levels and either initiate *tup* (DA2/AMP) or *col* (DA3/DO5) feed-forward positive loops. Positive auto-regulation, a hallmark of bistable systems (*Graham et al., 2010*; *Park et al., 2012*), of either *tup* or *col* distinguishes between DA2 and DA3 identities (*Boukhatmi et al., 2012*; *Figure 4L*).

## *eya* and *so* act sequentially in specifying DA3 and DO5 identity

*Drosophila eya* gives rise to 3 different mRNA and protein isoforms (*eya-RA, eya-RB, eya-RC*) corresponding to the alternate use of different Transcription Start Sites (TSS) (Flybase FBgn0000320; *Figure 5—figure supplement 1*). FISH performed with a probe complementary to a common coding region, $eya^e$, detected *eya* expression in the Col PMC, DA2/AMP, DA3/DO5 and LL1/DO4 PCs and DO5 lineage (*Figure 5B–E,O* and *Figure 5— figure supplement 2*). However, an intronic probe specific for transcripts initiated from the 5'-most TSS ($eya-RB^i$; *Figure 5—figure supplement 1*) revealed that *eya-RB* transcription was restricted to the AMP, DO5 FC/muscle and the LL1/DO4 PC (*Figure 5F–I,P* and *Figure 5— figure supplement 2*), implying that, conversely, *eya-RA/RC* is specifically expressed in the PMC, the DA2/AMP and the DA3/DO5 PCs. This indicated a temporal shift in TSS, leading to sequential production of different Eya protein isoforms. In *eya* mutants, *col* transcription was detected in PMC cells like in wt (not shown), but prematurely lost in the DA3/DO5 PC, and undetectable in the DA3 FC (*Figure 5R,S*), showing that *eya-RA/RC* is required for sustained *col* transcription during PC specification. Reciprocally, both *eya-RA/RC* transcripts in the DA3/DO5 PC (*de Taffin et al., 2015*) and *eya-RB* transcripts in the DO5 FC (*Figure 5T,U*) were lost in *col* mutants, revealing that Eya and Col positively regulate each other transcription. Whether, both *eya-RA/RC*, and *eya-RB* are under direct control of Col binding to a dedicated cis-regulatory module (CRM) (*de Taffin et al., 2015*; *Figure 5—figure supplement 1*), or *eya-RB* control is indirect, and requires prior expression of *eya-RA/RC* at the PC stage (*Figure 5X*), remains unknown.

Eya protein phosphatases interact with Six family TFs, Six4, Optix (Op) and So in *Drosophila* (see introduction). We found that *so* was transcribed in the DA2/AMP, DA3/DO5 and LL1/DO4 PCs and subsequently maintained only in the DO5 FC and muscle (*Figure 5J–M, Q* and *Figure 5— figure supplement 2*). *col* transcription in DL PCs was normal in *so* mutants, showing that So is not required for *Eya-RA/RC* regulation of *col* transcription (not shown). On the contrary, *col* was ectopically transcribed in the DO5 FC (*Figure 5V,W*), showing that *so* activity contributes to repress *col* transcription in this lineage. *so* transcription in the DO5 FC and contribution to distinguishing between the DA3 and DO5 identities suggests that *so* could act downstream of N in this process (*Crozatier and Vincent, 1999*). The difference between the *eya* and *so* DA3 mutant phenotypes (*Figure 2D,F*, *Figure 2—figure supplement 2*, and *Videos 4* and *6*) suggested that *eya* was partnering with *Six4* to positively regulate *col*. To verify this assertion, we analyzed DA3 Col expression in *Six4* mutants. The loss of Col expression (*Figure 5—figure supplement 3*), very similar to that observed in *eya* mutants (*Figure 2D*), supports the conclusion that Eya partners with Six4 to positively, and with So to negatively regulate *col*. Thus, Six4 and So play distinct roles during muscle specification (*Figure 5X*).

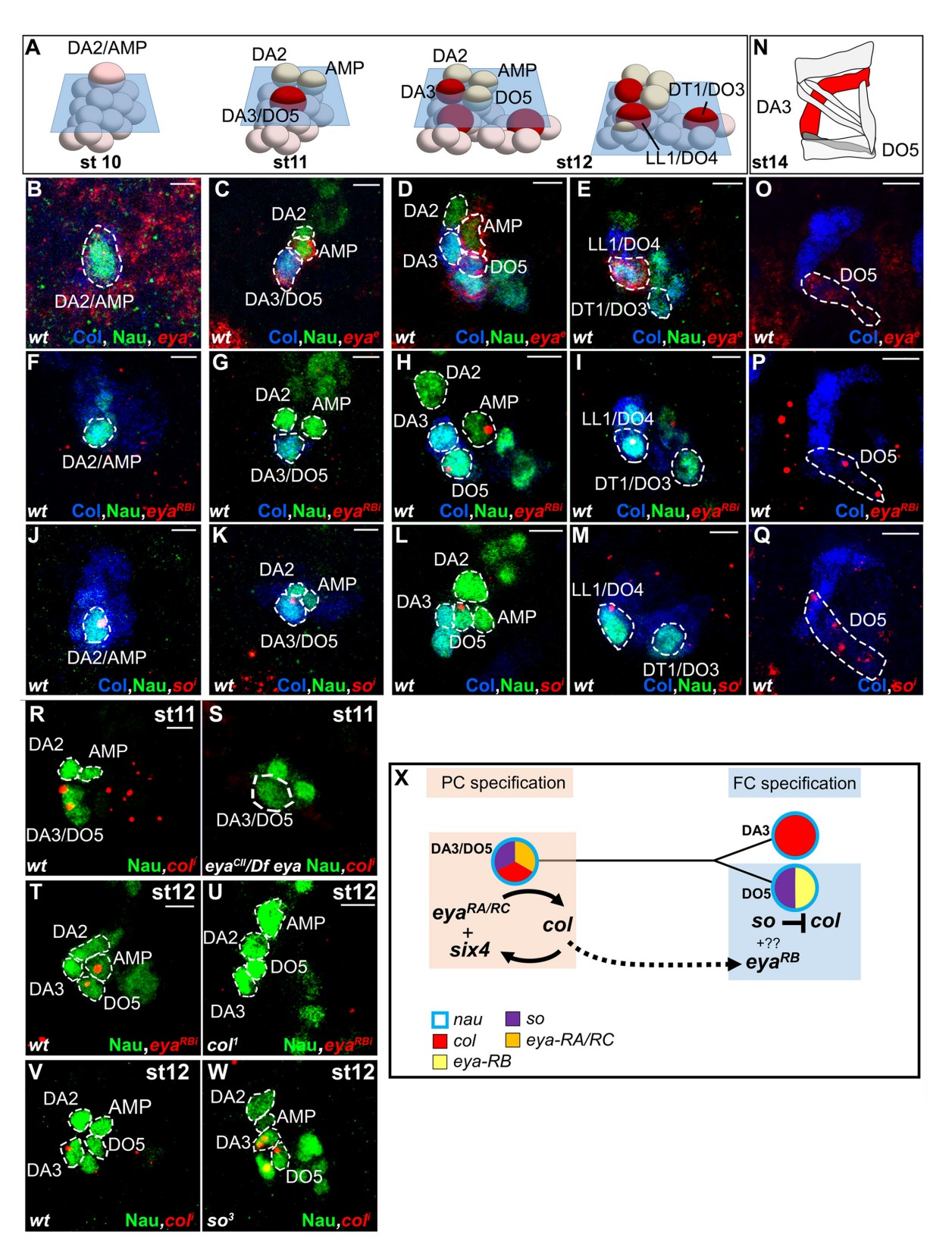

**Figure 5.** Sequential *eya* and *so* transcription and control of *col* transcription in distinct muscle lineages. (A) Schematic representation of the positions of DL PCs and FCs in stage 10, 11 and 12 wt embryos, reproduced from *Figure 3A*; the blue trapeziums indicate the planes of section shown below,
*Figure 5 continued on next page*

*Figure 5 continued*

panels (**B–M**). (**B–M** and **O–Q**) ISH to *eya* and *so* transcripts (red) in wt embryos stained for Col (blue) and Nau (green). (**B–E**), *eya* expression in the DA2/AMP (**B**), DA3/DO5 (**C**) AMP and DO5 FC (**D**) LL1/DO3 PC (**E**). (**N**) Schematic representation of the DL muscle pattern, DA3 in red and DO5 in grey. (**O**) DO5 *eya* expression. (**F–I**) *eya*-RB transcription in the AMP DO5 FC (**H**) and LL1/DO4 PC (**I**). (**P**) DO5 *eya*-RB transcription. (**J–M, Q**) *so* transcription in the DA2/AMP, DA3/DO5 and LL1/DO4 PCs, DO5 FC and muscle. (**R,S**) Loss of *col* transcription in the DA3/DO5 PC in *eya* mutants. (**T–U**) Loss of *eya*-RB transcription in *col* mutants (**U**). (**V,W**) *col* ectopic transcription in the DO5 FC in *so* mutants. Embryos in R-W are co-stained for Nau (green). (**X**) Summary diagram of *eya* and *so* expression and function in DL muscle lineages. Bars: 5 μm

The following figure supplements are available for figure 5:

**Figure supplement 1.** Schematic representation of the *eya* genomic region and transcripts.

**Figure supplement 2.** Sequential *eya* and *so* transcription.

**Figure supplement 3.** Loss of DA3 Col expression in *Six4* mutant embryos.

## Positional information; *noc* distinguishes between DA3 and DT1 identity

*noc* is required for DA3 Col expression and DA3 formation (*Figure 2*). *noc* and its paralog *elbow (elb)*, act as transcription repressors in morphogenesis of appendages, tracheal branches and specification of monochromatic receptors in the retina (*Dorfman et al., 2002*; *Nakamura et al., 2004*; *Wernet et al., 2014*) but a function in myogenesis was not reported. A deficiency removing *elb (Df (2R)exel6035)* had no DL muscle phenotype, while the phenotypes of *noc*[35ba2] mutants and a deficiency removing both *elb* and *noc* were identical (*Figure 1—source data 1* and *Figure 2E*). Thus, only *noc* is required for DL muscle specification. FISH experiments revealed *noc* transcription in Col PMC cells which express low Nau level, followed by the DA2/AMP, the DA3/DO5 and the LL1/DO4, but not the DT1/DO3 PC (*Figure 6B–E*). In *noc* mutants, *col* transcription was detected in the PMC and DA3/DO5 PC (not shown) but completely lost from the DA3 FC (*Figure 6F,G*), correlating with the loss of Col DA3 expression after the PC stage (*Figure 2E,K*). The DA3>DT1 transformation observed at stage 15 in *noc* mutants (*Figure 2Q*) was therefore intriguing, since a DA3>DA2 identity shift was observed in other mutants where DA3 Col expression was lost, namely *col, eya* and *Six4* (*Figure 2D* and *Figure 5—figure supplement 3*). Since DT1 identity requires S59 expression in the DT1 FC (*Knirr et al., 1999*), we analyzed *S59* expression in *noc* mutant embryos and found that it was ectopically expressed in the DA3 FC (*Figure 6H,I*), consistent with DA3>DT1 transformation. This finding suggested that loss of DA3 Col expression in *noc* embryos was secondary to gain of S59 expression. To test this possibility, we expressed S59 in all myoblasts, using the Twist-Gal4 pan-mesodermal driver. While the muscle pattern was severely disorganized, loss of Col expression in Twi>S59 embryos demonstrated S59 ability to repress *col* mesodermal expression (*Figure 6—figure supplement 1A,B*). We next analyzed *col* transcription in S59 mutants and found that it was ectopically transcribed in one posterior DL FC, likely DT1 (*Figure 6—figure supplement 1C,D*). On the one hand, these data confirmed that S59 represses *col* transcription in the DT1 lineage, via an incoherent feed-forward loop initiated by Col activation of *S59* in the DT1/DO3 PC (*Enriquez et al., 2012*). On the other hand, *noc* and *S59* loss-of-function and *S59* gain-of-function data revealed a double negative regulatory cascade where *noc* repression of *S59* maintains *col* transcription in the DA3 lineage and DA3 identity (*Figure 6J*). Noc expression in the DA3/DO5 PC and not the DT1/DO3 PC (*Figure 6E*) thus distinguishes between DA3 and DT1 identities (*Figure 7*).

Analysis of the DL muscle phenotypes showed that formation of the LL1 muscle is also affected in *noc, aop* and *eya* mutants (*Figure 2K,W* and *Figure 2—source data 1*). Kr was previously shown to be expressed in the LL1 PC and required for LL1 development (*Ruiz-Gomez et al., 1997*). We found that Kr expression in the LL1/DO4 (but not DA1) PC required both *noc* and *eya* activity, but neither *aop* nor *col* (*Figure 6—figure supplement 2*) (*Enriquez et al., 2012*). These data further underline the intricate wiring and combinatorial nature of transcriptional regulations specifying muscle identities (*Figure 7*).

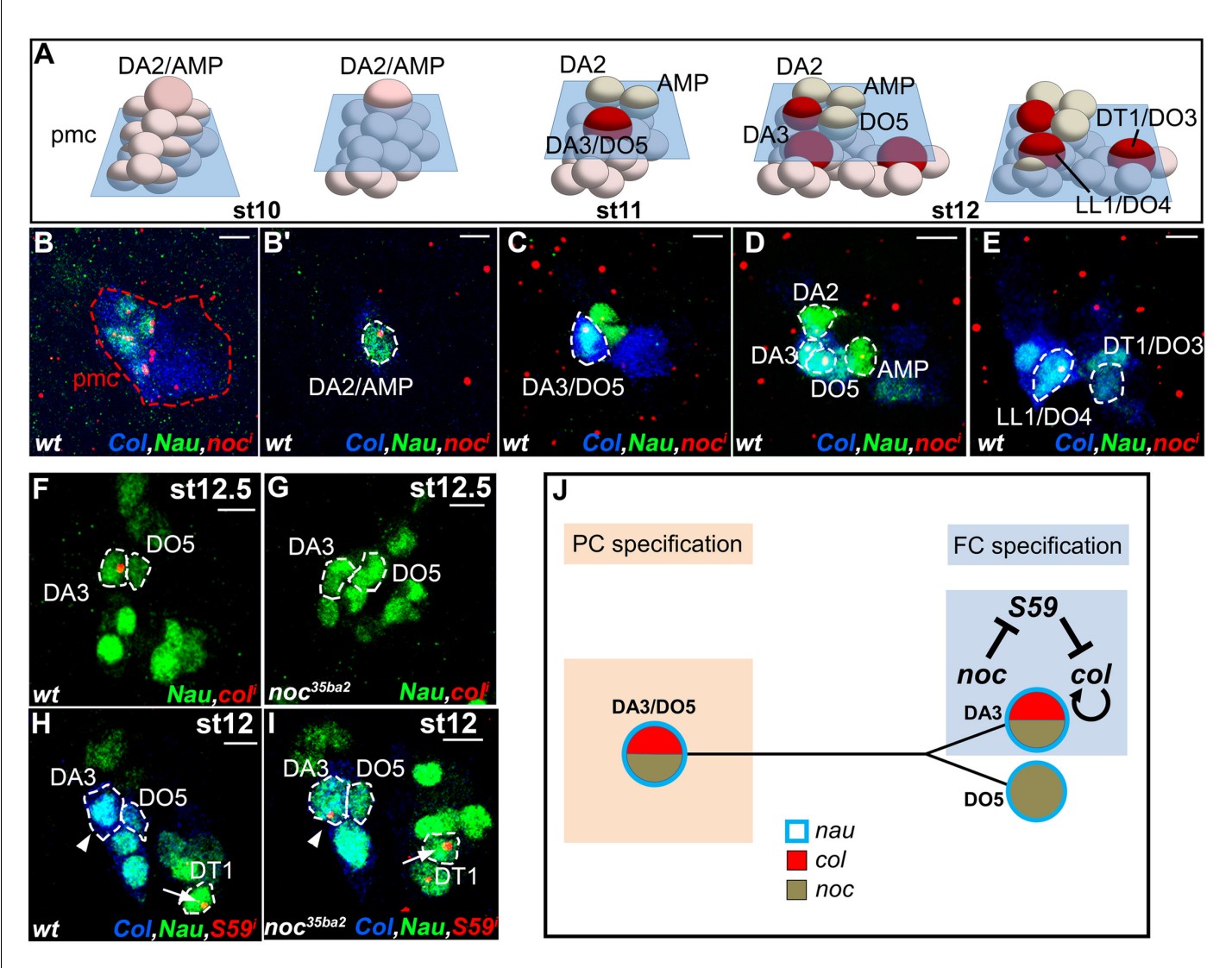

**Figure 6.** *noc* transcription and control of *col* and *S59* transcription in DL muscle lineages. (**A**) Relative positions of DL PCs and FCs between stages 10 and 12, reproduced from **Figure 3A**; the blue trapeziums indicate the planes of section shown below, panels (**B–E**). (**B–E**), Embryos co-stained for Nau (green) and Col (blue); (**B**) *noc* transcription (red) in a small subset of Col PMC cells expressing low Nau level (**B**), the DA2/AMP (**B'**), DA3/DO5 and DA3 and DO5 FCs (**C,D**) and LL1/DO4 but not the DT1/DO3 PC (**E**). (**F–I**) Embryos stained for Nau (green) and (**H,I**) Col (blue). (**F, G**) loss of *col* transcription in the DA3 FC (red dots) in noc mutants. (**H**) wt *S59* transcription in DT1 (red dots, arrow) and (**I**) ectopic transcription in the DA3 FC (arrowhead) in noc mutant embryos. (**J**) Summary diagram of noc expression and function in DL muscle lineages. Bars: 5 μm

The following figure supplements are available for figure 6:

**Figure supplement 1.** S59 represses *col* transcription.

**Figure supplement 2.** *eya* and *noc* are required for Kr expression in the LL1 FC.

## Discussion

A 3-steps model of muscle identity specification has been put forward 18 years ago in *Drosophila*, based on a limited repertoire of muscle iTFs (**Carmena et al., 1998**). In this work, we took an unbiased genetic approach and focused on a small group of muscles, to identify new transcription regulators of muscle identity. By combining functional and expression analyses of these new iTFs at the

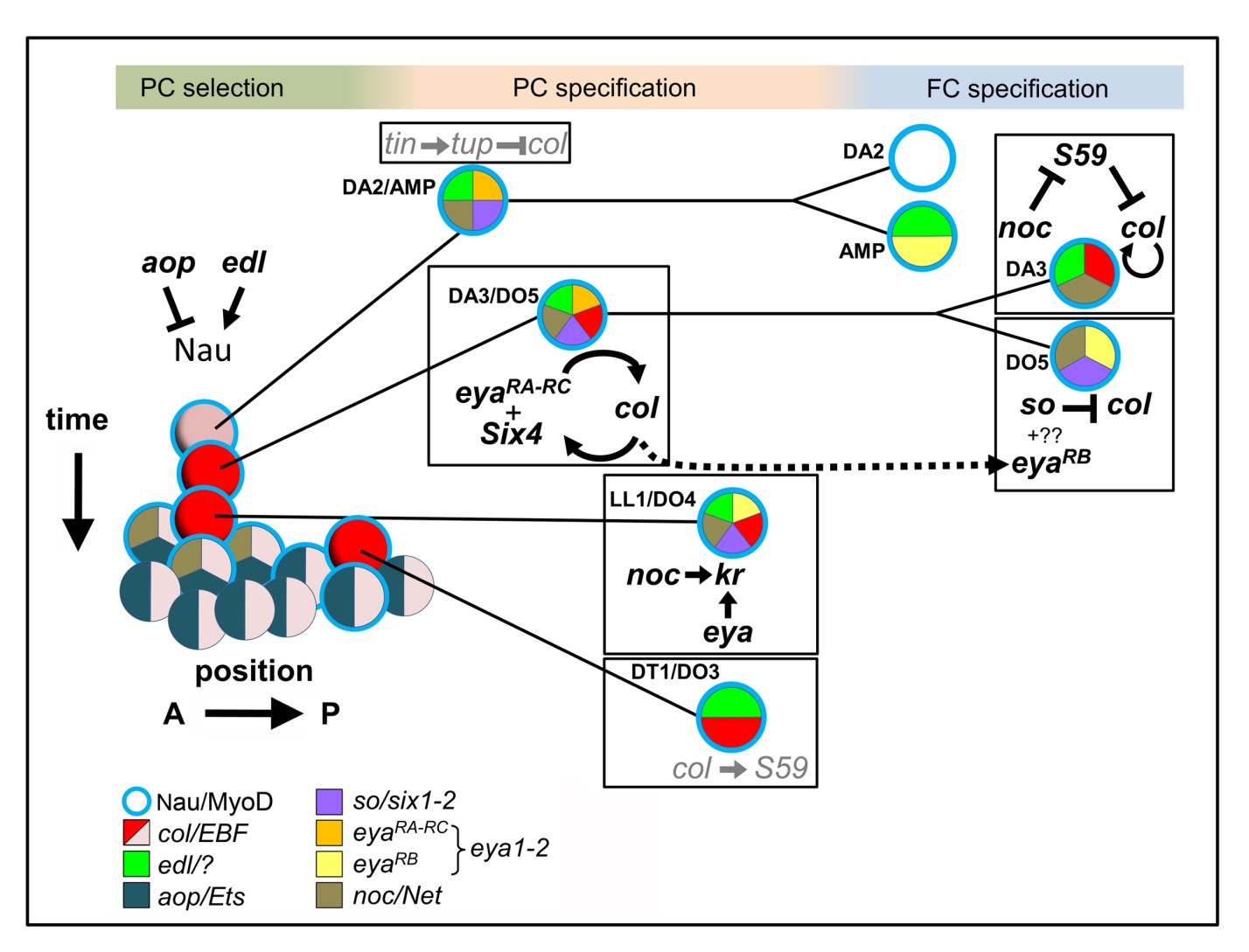

**Figure 7.** Intertwined transcriptional control of DL muscle identity: A progressive resolution of the possible. Diagrammatic representation of transcription regulatory interactions and loops operating in DL muscle identity specification. One abdominal segment is considered. 3 steps are indicated on top and color-shaded. Left, PC selection; the A/P axis is on the abscissa and the developmental time on the ordinate. *aop* and *edl* positively and negative regulate Nau expression (blue circle) in a subset of PMC cells, favoring and inhibiting selection of PCs from PMC cells, respectively. Center, PC specification: the dorsal DA2/AMP and dorso-lateral DA3/DO5, LL1/DO4 and DT1/DO3 PCs are represented. Right, FC specification; only the DA3 and DO5 lineages are detailed. Color coding of expression of the different iTFs, and the names of their vertebrate orthologs are indicated. Genes and either positive (arrow), or negative (crossed line) regulatory steps identified in this study are drawn in black; previously reported interactions are in grey.

primary transcripts level, with previous data, we propose a novel, dynamic view of the transcriptional control of muscle identity by evolutionarily conserved TFs.

## Noc, a new iTF responding to multimodal information

Noc and Elb are two transcription repressors belonging to the NET family of C2H2 zinc finger proteins (*Dorfman et al., 2002*; *Nakamura et al., 2004*). Until now, identification of a Tin-dependent mesodermal *noc* enhancer was the only suggestion of a possible function in muscle development (*Jin et al., 2013*). We show here that *noc* is required for distinguishing between DA3 and DT1 muscle identities. Primary transcript analyses revealed that *noc* maintenance of *col* transcription in the DA3 lineage involved a double Noc —|S59 —|col negative loop. Col activation of S59 expression

(*Enriquez et al., 2012*) initiates a negative, 'incoherent' feed-forward loop in the DT1/DO3 PC, resulting in Col repression and DT1 identity. Noc breaks this loop in the DA3 lineage (*Figure 7*). *noc* is transcribed in myoblasts expressing low Nau level whose number is controlled by EGF-R signaling, and selected PCs (*Figure 6* and *Figure 3—figure supplement 2*). However, it is not transcribed in the DT1/DO3 PC which is specified under control of abdominal Hox proteins (*Enriquez et al., 2010*). Thus *noc* regulation integrates multimodal signaling and positional inputs. Functions of vertebrate NET proteins Nolz-1/Nolz2 and ZNF503/nlz1 have, so far, been addressed in the central nervous system. Yet, zebrafish *Znf503* and mouse *nolz1* are expressed in branchial arches (zfin ID: ZDB-Gene-031113-5, genebank accession number NM-198840; *Chang et al., 2013*). Whether this expression corresponds to migrating myoblasts which contribute to facial muscle development and express orthologs of other iTFs studied here (*Nathan et al., 2008*; *Sambasivan et al., 2009*; *Tolkin and Christiaen, 2012*; *Razy-Krajka et al., 2014*; *Diogo et al., 2015*), remains to be explored.

## Temporal control of muscle PC selection

Our previous finding that several PCs are sequentially selected from the Col PMC and each express a specific iTF code brought to light the importance of time in muscle identity specification (*Boukhatmi et al., 2012*). We report here that *aop* and *edl*, are required for the robustness of sequential selection of the DA2/AMP and DA3/DO5 PCs. A consequence of lack of Aop function is the premature, concomitant selection of several Nau-expressing PCs, when sequential in wt embryos. These data indicate that Aop acts to restrict EGF-R MAPK activity to prospective PCs, as previously suggested (*Carmena et al., 2002*). Inversely, *edl* expression in selected cells and delayed PC delamination in *edl* mutant embryos indicate that Edl ensures timely progression of primed cells to a stable PC fate. Thus, proper regulation of EGF-R signaling effectors is both required for the proper number and temporal sequence of PC selections. During eye development, the differentiation of all ommatidial cell types is triggered in a stereotypical sequence by reiterated EGFR use (*Freeman, 1996*). Likewise, EGF-R signalling regulates pulses of cell delamination from the ectoderm (*Brodu et al., 2004*). Whether pulses of EGF-R signalling control serial PC selection remains to be determined. Parallels between serial PC selection and successive waves of neuroblast (NB) selection (*Doe, 1992*; *Berger et al., 2001*) are striking. However, in NB lineages, the focus of investigation has shifted to the sequential expression of Temporal Transcription Factors (TTFs) in each NB progeny, which specifies the temporal identity of neurons (*Isshiki et al., 2001*), and how birth time controls NB identity has not been determined.

## Nau expression in early steps of myogenesis. A new Nau function?

Our observation of an increased number of Nau-positive cells in *aop* mutants revealed that Nau/MyoD is expressed in myoblasts primed to a PC fate. Nau role in generic myogenesis versus identity aspects has been debated (*Balagopalan et al., 2001*; *Wei et al., 2007*). We have previously proposed that Nau iTF functions could reflect lineage-dependent cooperation with other iTFs (*Enriquez et al., 2012*). Another b-HLH protein, Lethal of Scute (L(1)sc), has been proposed to act as a promuscular gene (*Carmena et al., 1995*), based on its pattern of expression and regulation in PMCs and PCs, and by analogy to the role of proneural b-HLH proteins in NB selection. Yet, only minor muscle patterning defects were observed in *l(1)sc* mutants (*Carmena et al., 1995*). The detection of Nau expression in PMC cells subject to high EGF-R signaling raises the possibility that Nau could play earlier functions than previously thought in the PC specification process.

## Translation of developmental time into muscle identity

The time lag between DA2/AMP and DA3/DO5 PC emergence coincides with dorsal regression of Tin expression, due to an auto-regulatory circuit in which Tin progressively limits its own transcription (*Johnson et al., 2011*). We previously showed that only the DA2/AMP PC inherited Tin (*Boukhatmi et al., 2012*). Considering Tin levels as a translation of developmental time, our new results show that EGF-R control of serial PC selection converts this translation into sharp transcriptional decisions and ultimately distinct muscle identities (*Figures 4* and *7*). Tup and Col direct upregulation of their own transcription after the PC stage (*Enriquez et al., 2010*; *Boukhatmi et al., 2012*, *2014*) can explain why small differences in initial expression levels are transformed into stable

muscle fates. Our data thus provide a new paradigm for how birth timing controls cell identity during development (*Figure 4L*).

### Different Eya isoforms and Six partners are sequentially involved in muscle development

*Drosophila* Six proteins, Six4, So and Op correspond to vertebrate Six1/2, Six4/5 and Six3/6, respectively (*Kenyon et al., 2005*). Six1/2/4/5 proteins have been reported to interact with *eya* in controlling the myogenic progenitor cell population in mouse, while no role was found for Six3/6 in this process (*Relaix et al., 2013*). Likewise, we did not observe DL muscle pattern defects in embryos lacking *op (Df(2R)Exel6055)*. *Drosophila Six4* was previously proposed to interact with Eya in regulating somatic muscle development, both genes showing similar expression pattern (*Clark et al., 2006*; *Liu et al., 2009*). Tin in vivo binding to the *so* locus (*Liu et al., 2009*; *Jin et al., 2013*) raised the possibility that *so* could also be involved in muscle development. Interestingly, temporal ChIP profiles indicated Tin binding to *six4* earlier than to *so* (*Jin et al., 2013*), suggesting sequential regulation. Our data show that one *eya* isoform, *eya-RB*, is transcribed later than the other isoforms (*eya-RA/ RC*, *Figure 5—figure supplement 1*), owing to a switch in TSS, correlating with the profile of in vivo RNA polymerase II binding (*Bonn et al., 2012*). Together, these expression data and our finding that *eya* and *Six4* regulate positively, and *so* negatively, *col* transcription, suggest that *Eya* could switch from activator to repressor, by changing partner, from Six4 to So. We thus hypothesize that sequential partnering of different Eya isoforms and Six proteins could contribute the diversity of iTF codes and muscle morphologies (*Figure 7*). Various contributions of mouse Six1/2/4 to the Pax3/ MyoD transcription regulatory network controlling early myogenesis in different embryonic territories have recently been reported (*Relaix et al., 2013*). It would be interesting to determine whether different *eya* isoforms are also involved in different Six partnerships and control different facets of muscle development in vertebrates.

### An integrated view of the transcriptional control of muscle identity

Characterisation of *aop, edl, eya, noc* and *so* functions and transcription dynamics revealed that distinguishing between DA3, versus DA2, DT1 or DO5 identities involves specific sequences of transcriptional regulations integrating temporal and positional cues (*Figure 7*). The functions of these, and previously characterized iTFs, underline the intricacy of positive and negative regulatory loops acting at successive steps in different muscle lineages (*Figure 7*). Besides clear transformations suggestive of complete identity switch, a significant fraction of muscles show incomplete transformations in iTF mutant embryos. This supports the idea that, rather than lineage-specific 'master iTFs', stereotypy of Drosophila muscle patterns relies upon combinatorial inputs of multiple iTFs during PC and FC specification. The finding that TFs combinatorially specify muscle identity (*Enriquez et al., 2012*; *de Joussineau et al., 2012*; *Boukhatmi et al., 2014*, this report) indicates that activation of their target genes is context-dependent and involves multiple cis-regulatory elements. Multiple levels of cross-regulation (*Figure 7*) could provide robustness to the final muscle pattern.

While a function of Nolz proteins in the mesoderm remains to be investigated, Nkx2.5, Eya, Six1, Islet1, Col/Ebf and MyoD, are core components of transcriptional regulatory networks controlling the development of pharyngeal/facial muscles originating from the cardio-pharyngeal territory in chordates. Our data raise the possibility that these conserved mesodermal TFs could combinatorically control muscle regional diversity in vertebrates, attested by human muscular dystrophies (*Emery, 2002*). but whose molecular basis remains poorly understood. It is reasonable to speculate that these TFs have been co-opted in different wirings during evolution to generate the muscle lineage diversity found in the animal kingdom.

## Materials and methods

### Screening procedure and mutant alleles

Deficiency lines from the Bloomington Drosophila Stock Center were balanced over a CyO–*wg*–*LacZ* chromosome to genotype embryos (*Chanut-Delalande et al., 2014*). Screening for embryonic phenotypes was as in *Chanut-Delalande et al. (2014)*, except that embryos were stained with a monoclonal mouse antibody against the Col protein (*Dubois et al., 2007*). Genetic complementation

assays were used to identify genes in chromosomal deficiencies whose loss led to DA3 phenotype. Homozygous $aop^1$, $edl^{L19}$, $eya^{CII/IID}$, $so^3$, $noc^{35ba2}$, $salm^1$, and $col^1$ homozygous mutants showed muscle phenotypes identical to trans-heterozygous mutants over deficiency. They were thus considered as null alleles and consistently used for phenotypic analyses. For eya analysis, we used $eya^{CII/IID}/Df(2l)BSC354$ trans-heterozygous embryos as $eya^{CII/IID}$ homozygous embryos present a strong myoblast fusion defect (not shown), not observed in transheterozygous and probably due to a secondary mutations on the $eya^{CII/IID}$ chromosome. Col::moeGFP expression under control of a late mesodermal col CRM ($col^{LCRM}$; previously named 2.6_0.9c; *Dubois et al., 2007*) was used to visualize the DA3 muscle contours in mutant embryos. The col PMC cells and PCs were visualized in col mutant embryos by LacZ expression under control of the early mesodermal col CRM, $col^{ECRM}$ (previously CRM276; *Enriquez et al., 2010*). For all deficiency screening, sample size was estimated empirically (>100 stage 14–16 embryos, in duplicates). For aop, edl, eya, so, and noc, mutant analyses, sample sizes are indicated in the text and the legend of *Figure 2—source data 1*.

## Immunohistochemistry, in situ hybridization and imaging

Antibody staining and in situ hybridization with intronic probes were as described previously (*Dubois et al., 2007*). Primary antibodies were: mouse βPS integrin, anti-Col (*Dubois et al., 2007*), anti-GFP (Torrey Pines Biolabs), anti-β-galactosidase (Promega, Madison, Wisconsin), rabbit anti-Tin (Manfred Frasch, Erlangen, Germany), anti-Nau (Bruce Paterson, Bethesda, USA), anti-Kr (Ralf Pflanz, Goettingen, Germany), anti-β3-tubulin (Renate Renkawitz-Pohl, Marburg, Germany). Secondary antibodies were: Alexa Fluor 488- and 555-conjugated antibodies (Molecular Probes), and biotinylated goat anti-mouse (Vector Laboratories). Digoxygenin-labelled antisense RNA probes were transcribed in vitro from PCR-amplified DNA sequences, using T7 polymerase (Roche Digoxigenin labelling Kit). For aop, edl and eya-RB, 3 non overlapping 600 nucleotide (nt) probes were pooled together; for noc, a single probe spanning the entire 302 bp intron; for so, a 2771nt probe hydrolyzed to ~600nt fragments (Cox et al., 1984). The primer pairs used to amplify the different intron fragments are listed below, with the T7 promoter indicated by small characters.

 *aop1*: CTCATTGTATGCACGGTACG
 *aop1*T7: ccgaattctaatacgactcactatagggATAGCTGCGGCAGAAGCAGG
 *aop2*: GCAACAGCAACACTCCAATC
 *aop2*T7: ccgaattctaatacgactcactatagggAGACGGTGCGGGCAGAAATTGGG
 *aop3*: AAGAGAAAGAGCACGGCAAG
 *aop3*T7: ccgaattctaatacgactcactatagggAGATCGGCGACGTTCTCCGAGAC
 *edl1*: GGGAGGTGGAAATGACAAAC
 *edl1*T7: ccgaattctaatacgactcactatagggCATCGTCTGCCTGACGTCTG
 *edl2*: CCAAATATCGCCGATAAGCC
 *edl2*T7: ccgaattctaatacgactcactatagggAGACTGCGCACAGGATGCACACC
 *edl3*: GAAGATCGACCAGACTTAGG
 *edl3*T7: ccgaattctaatacgactcactatagggAGAAGCGGCGTCGAGATTCCCAG
 *eyaRB1*: GTTCCTCTAGCTCCGAAATG
 *eyaRB1*T7: ccgaattctaatacgactcactatagggTTACGCCGGAGTTGTGAGGG
 *eyaRB2*: GACAGCATCGGAGACAACAC
 *eyaRB2*T7: ccgaattctaatacgactcactatagggCCCGGCCACAAACGAGAAAC
 *eyaRB3*: AGCCCAGTCAAATGCGAAAC
 *eyaRB3*T7: ccgaattctaatacgactcactatagggATGCGTGTCCGTGTCGCTAC
 *noc1*: CGACGGTTAGTATTGACTAAG
 *noc1*T7: ccgaattctaatacgactcactatagggGGCGTCCATCTGTTATGAATAAAATG
 *so1*: TCCACGTTTCCAAGTTGGCTACTC
 *so1*T7: ccgaattctaatacgactcactatagggAATGCGGCATGTTCGATGCTCGATAATCGG

Confocal sections were acquired on Leica SP5 or SPE microscopes at 40× magnification, 1024/1024 pixel resolution. Images were assembled using ImageJ and Photoshop softwares. 3-D reconstructions of the topology of DL PCs and FCs were made from optimized section, 'using volocity (PerkinElmer) or Imaris (Bitplane) Softwares'. Images presented are representative of observations of at least 10 embryos per genotype at a given stage and between five and six segments per embryo. To compare the size of the Col and Nau expression domains in wt and aop mutants, optimized

stacks of double-stained embryos in the same orientation were flattened, and the largest diameter of each domain measured. Statistical analysis was with GraphPad Prism5 using unpaired t-test.

## Acknowledgements

We thank the Bloomington Stock Center and colleagues for *Drosophila* strains and Manfred Frasch, Bruce Paterson, Ralf Pflanz, Renate Renkawitz-Pohl for antibodies. We acknowledge Laetitia Bataillé, Hadi Boukhatmi, Véronique Brodu, Alice Davy and Jean-Antoine Lepesant for hepful discussion and critical reading of the manuscript and the help of Brice Ronsin, Toulouse RIO Imaging platform, and Julien Favier for maintenance of fly stocks.

## Additional information

### Funding

| Funder | Grant reference number | Author |
| --- | --- | --- |
| Centre National de la Recherche Scientifique | | Laurence Dubois<br>Jean-Louis Frendo<br>Hélène Chanut<br>Michèle Crozatier<br>Alain Vincent |
| Ministère de l'Éducation Nationale | | Laurence Dubois<br>Jean-Louis Frendo<br>Hélène Chanut<br>Michèle Crozatier<br>Alain Vincent |
| AFM-Téléthon | 14895-SR | Laurence Dubois<br>Jean-Louis Frendo |
| Agence Nationale de la Recherche | 13-BSVE2-0010 | Alain Vincent |

The funders had no role in study design, data collection and interpretation, or the decision to submit the work for publication.

### Author contributions

LD, Conception and design, Acquisition of data, Analysis and interpretation of data, Drafting or revising the article; J-LF, Acquisition of data, Analysis and interpretation of data, Drafting or revising the article; HC-D, Acquisition of data, Analysis and interpretation of data, Contributed unpublished essential data or reagents; MC, Analysis and interpretation of data, Drafting or revising the article, Contributed unpublished essential data or reagents; AV, Conception and design, Analysis and interpretation of data, Drafting or revising the article

### Author ORCIDs

Jean-Louis Frendo, http://orcid.org/0000-0003-0118-5556
Alain Vincent, http://orcid.org/0000-0002-2769-7501

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
