## [Decision Letter]

Thank you for submitting your article "Genetic dissection of the Transcription Factor code controlling serial specification of muscle identities in *Drosophila*" for consideration by *eLife*. Your article has been reviewed by three peer reviewers, and the evaluation has been overseen by a Reviewing Editor and K VijayRaghavan as the Senior Editor.

The reviewers have discussed the reviews with one another and the Reviewing Editor has drafted this decision to help you prepare a revised submission.

The identity of muscle progenitors and their founder cell progeny is specified by a combinatorial code of expression of transcription factors. In this work, the authors focus on the Collier expressing DA3 muscle in *Drosophila*. By genetic analysis, they identify known genes for which a role in muscle development has not yet been described and expand our knowledge on the role of genes already known to affect muscles. New roles for the genes Anterior Open, ETS domain lacking, Eyes absent, No Ocelli, and Sine oculis are demonstrated in progenitor and founder cell specification. The authors examine muscle perturbations and changes in expression of specific markers to delineate the precise decisions involved in muscle specification. This manuscript provides the most complete example to date of this cascade of events and reveals the involvement of dynamic feedback loops. Furthermore, it presents the novel finding that the timing of progenitor cell specification is critical to its subsequent development. Overall, this is important work that demonstrates the dynamic nature of the muscle identity specification process and its sequential nature.

Prior to acceptance, the following points should be addressed:

Major revisions

1) Quantification and 3D data are necessary for mutants (Figure 2). Some information is provided on the frequency of the defects (e.g. transformations) but could be expanded and should be placed in a table. Some transformations are very hard to see (2H), and might be aided by viewing the data in 3D.

2) 3D data for Figure 4 would also be beneficial, as it is very hard to interpret the Col pattern in the mutants from the data provided.

3) The FISH for *eyaRB* and so are not convincing. This is an important point for the manuscript since expression of these genes in muscle progenitors is the only evidence consistent with an autonomous role.

4) Eya and, in particular, *aop* mutants exhibit defects in many muscles. What is the evidence that these defects are autonomous in the mesoderm and/or in DA3 specifically? Can DA3 be rescued by expression of these genes within the mesoderm? Though unlikely to work, have the authors tried knockdown of any of these genes in the mesoderm?

5) To show the Noc —| *S59* —|*col* double negative circuit/gate, the authors should attempt to rescue the *noc* phenotype with a *S59* loss-of-function; not just mimic the *noc* phenotype with a *S59* gain-of-function, which is consistent with the model, but not an unequivocal demonstration.

6) While it may be beyond the scope, changes in the expression profiles of markers (e.g. for DA2) would provide much stronger evidence of DA2-DA3 transformations than subjective studies using the shape to infer an autonomous/intrinsic change.

7) In general, the authors could be more explicit about the possible impact of the *aop* and *edl* mutations on EGFR signaling. Placing the phenotypes in the context of loss- of gain-of- EGFR signaling would be informative if this is possible in the time frame of the revisions.

In the model shown in Figure 7, adding EGFR would be helpful, including the possible impacts of *aop* and *edl*. This is also true for Figure 4. It seems that *aop* could act to restrict EGFR-MAPK activity in the prospective DA2/AMP progenitor cells, whereas *edl* could be necessary for proper EGFR-MAPK, but this is not clear in the summary.

8) The paper is generally a bit difficult to read for non-specialists of muscle development in *Drosophila*, but the summary diagrams inserted in various figures help a lot. In particular, it is difficult to understand how the authors identified the genes starting from the deficiencies; a simple explanation would help the non-specialist.

---

## [Author Response]

*Major revisions*

*1) Quantification and 3D data are necessary for mutants (Figure 2). Some information is provided on the frequency of the defects (e.g. transformations) but could be expanded and should be placed in a table. Some transformations are very hard to see (2H), and might be aided by viewing the data in 3D.*

Following the referees’ request, 3D views of *aop, edl*, eya, *noc* and so mutant embryos have been added (Video 1 to 6, and associated Figure 2—figure supplement 2). A table describing in more detail the frequency of the defects observed in the different mutants has also been added ([Supplementary-material SD2-data] which replaces former Figure 2—figure supplement 2).

*2) 3D data for Figure 4 would also be beneficial, as it is very hard to interpret the Col pattern in the mutants from the data provided.*

We agree that 3D visualization of Col and Tin expression would be optimal. However, the surrounding of (low-level) Col expressing PCs by Tin expressing cells makes it very hard to properly visualize them in 3D. We think that the 2D figures shown in Figure 4 clearly illustrate the abnormal timing of Col-expressing PC selection in either aop or edl mutants. To re-enforce this point, we analyzed the pattern of tup transcription in aop and edl mutants (new panels J-K in Figure 4), which further strengthens our model, Figure 4 (see also answer to point 6).

*3) The FISH for eyaRB and so are not convincing. This is an important point for the manuscript since expression of these genes in muscle progenitors is the only evidence consistent with an autonomous role.*

We are not exactly sure of why our in situ data are “not convincing”. Figure 5 shows that *eya-RB* transcripts are detected in the DO5 FC (distinguishable from DA3 by its lack of Col) and the DO5 muscle Figure 5. Likewise, *so* transcription in the DA2/AMP, DA3/DO5, and LL1/DO4 PCs, and maintenance in the DO5 lineage, is clearly shown by Figure 5 and Q.

Nevertheless, to better address the reviewers’ concern, and better highlight the *eya* and *so* transcription patterns, we made a supplementary figure (Figure 5—figure supplement 2) showing only the *in situ* signal (red channel). The specific overlap of *eyaRB* and *so* transcription in the DO5 FC and muscle is consistent with the DO5 *so* phenotype and a cell autonomous role of EyaRB/So (see also answer to point 4). The early requirement of Eya in the DA3/DO5 PC precludes analysis of its specific role in the D05 lineage.

*4) Eya and, in particular, aop mutants exhibit defects in many muscles. What is the evidence that these defects are autonomous in the mesoderm and/or in DA3 specifically? Can DA3 be rescued by expression of these genes within the mesoderm? Though unlikely to work, have the authors tried knockdown of any of these genes in the mesoderm?*

We agree that identification of *aop* and *eya* mutants, using DA3 Col expression and muscle formation as a read-out, does not prove that *eya* and *aop* are cell-autonomously required in the DA3 muscle. However, cell autonomous function of *eya* and *aop* in somatic muscles is supported by previous genetic and molecular analyses (Halfon et al., 2000, Carmena et al., 2002, Liu et al., 2009; Ciglar et al., 2014). Since Aop is a transcription factor, both *aop* transcription in PMCs and the increased PMC expression of Nautilus which is observed in *aop* mutants (Figure 3) support a cell autonomous role in PC specification, a function previously suggested in dorsal Eve-expressing lineages (Carmena et al., 2002). Our present data showing the sequential modification of Nau expression in the DL PMC, and *col* transcription during the PC selection process in *aop* mutants add a dynamic dimension to previously published work. Because of its role in FCM versus PC identity (Carmena et al., 2002, our data) we suspect that pan>mesodermal expression of *aop* would not allow to follow rescue of individual muscles. It will certainly be rewarding to perform both *aop* gain of function and loss of function experiments when drivers allowing to specifically target each step in the PC selection process become available.

*eya* (Liu et al., 2009 and this paper) and *six4* (Clark et al., 2006) expression in the somatic mesoderm together with the *eya* mutant phenotype (Figure 2, Figure 5), and cross-regulation between *eya* and *col* in the DA3 PC (Figure 5 and de Taffin et al., 2015) all support the conclusion of a cell-autonomous role of Eya in the DA3 muscle. To support our assertion of an Eya/Six-4 cell autonomous role in DA3 specification, we examined DA3 Col expression in *Six4* mutant embryos. The data show a very similar phenotype to that observed in *eya* mutants with loss of DA3 Col expression in 84% of segments at stage 15, a loss already observed at the PC stage. The *six4* mutant DA3 phenotype is now shown in a new supplementary figure (Figure 5—figure supplement 3 and revised text, paragraph 2 subsection “*eya* and *so* act sequentially in specifying DA3 and DO5 identity”).

*5) To show the Noc —| S59 —|col double negative circuit/gate, the authors should attempt to rescue the noc phenotype with a S59 loss-of-function; not just mimic the noc phenotype with a S59 gain-of-function, which is consistent with the model, but not an unequivocal demonstration.*

Our model, Figure 6 and Figure 7, proposes that *noc* expression prevents S59 repression of *col* transcription in the DA3 lineage, a conclusion admitt*edl*y relying in part on S59 gain of function experiments. We could not succeed analyzing double *noc*/S59 mutant embryos in the imparted time. Therefore, in order to strengthen the Noc —| *S59* —|*col* double negative loop, we looked at *col* transcription in loss of function *S59* mutants. Ectopic col transcription was specifically observed in one abdominal DL FC where *noc* is not expressed (Figure 6), likely the DT1 FC (Knirr et al., Development 1999). This ectopic transcription, which confirms *col* repression by S59 in cells which do not express *noc* is shown in new panels C and D, in Figure 6—figure supplement 1, with accompanying text, paragraph 1 subsection “Positional information; *noc* distinguishes between DA3 and DT1 identity”.

*6) While it may be beyond the scope, changes in the expression profiles of markers (e.g. for DA2) would provide much stronger evidence of DA2-DA3 transformations than subjective studies using the shape to infer an autonomous/intrinsic change.*

Following the referees’ suggestion, we have analyzed *tup* transcription in the DA2/AMP and DA3/DO5 PCs in *aop* and *edl* mutant embryos, which predominantly show DA3>DA2 and DA2>DA3 transformations, respectively.

In wt embryos, *tup* is transcribed in the first-born PC selected from the *col*-expressing PMC, leading to *col* repression in this PC and DA2 identity. *tup* is not transcribed in the second-born PC, which maintains *col* transcription and adopts a DA3 identity (Boukhatmi et al., 2012). Our new data show a significant number of cases where both PCs transcribe *tup* in *aop* mutant embryos, while neither transcribes *tup* in *edl* mutants (New panels I-K in Figure 4 and text paragraph 2 subsection “*aop* and *edl* activities control sequential DA2/AMP and DA3/DO5 PC selection”). In addition to confirming a precocious selection of the DA3 PC in *aop* mutants, and delayed selection in *edl* mutants, the *tup* transcription data complement *col* transcription data (Figure 3) and provide further evidence that the DA3>DA2 and DA2>DA3 morphological transformations observed in *aop* and *edl* mutants, respectively, result from PC mis-specification due abnormal timing of selection.

*7) In general, the authors could be more explicit about the possible impact of the aop and edl mutations on EGFR signaling. Placing the phenotypes in the context of loss- of gain-of- EGFR signaling would be informative if this is possible in the time frame of the revisions.*

Complete loss of EGF-R signalling in star mutants, leads to loss of DL muscles (except for DT1; Figure 1—figure supplement 1), whereas supernumerary PCs form in Ras1Act mutants (Carmena et al., 1998), The aop and edl mutations display more subtle, identity phenotypes. We are therefore conscious that the impact of aop and edl mutations on timing and level of EGFR signaling deserves further investigation, but also feel that it falls beyond the scope of this manuscript.

*In the model shown in Figure 7, adding EGFR would be helpful, including the possible impacts of aop and edl. This is also true for Figure 4.*

In absence of more detailed investigation of EGF-R signaling (see above) we would prefer not to be more specific on the precise role of *aop* and *edl* in the different DL PCs in Figure 7 and Figure 4. In order for the scheme Figure 7, to better reflect new data, we added Nautilus regulation by *aop* and *edl* during the PC selection process.

*It seems that aop could act to restrict EGFR-MAPK activity in the prospective DA2/AMP progenitor cells, whereas edl could be necessary for proper EGFR-MAPK, but this is not clear in the summary.*

We have modified the text accordingly, (subsection “Temporal control of muscle PC selection”)

*8) The paper is generally a bit difficult to read for non-specialists of muscle development in Drosophila, but the summary diagrams inserted in various figures help a lot. In particular, it is difficult to understand how the authors identified the genes starting from the deficiencies; a simple explanation would help the non-specialist.*

Following the recommendation of the referees, we modified the text (paragraph 2 subsection “A genetic screen for muscle defects”) to better explain how we went from deficiencies to genes. We also tried our best to render the text less difficult to read for non-specialists of *Drosophila* muscle development and added a few more references to studies in vertebrates. We also modified the legend of summary model Figure 7.